# A Branch and Bound Framework for Stronger Adversarial Attacks of ReLU Networks

## Abstract

Strong adversarial attacks are important for evaluating the true robustness of deep neural networks. Most existing attacks find adversarial examples by searching the input space, e.g., using gradient descent, and may miss adversarial examples due to non-convexity. In this work, we search adversarial examples in the *activation space* of ReLU networks, which allows a *systematic search* of adversarial examples and can tackle hard instances where none of the existing adversarial attacks succeed. Unfortunately, searching the activation space typically relies on generic mixed integer programming (MIP) solvers and is limited to small networks and easy problem instances. To improve scalability and practicability, we use branch and bound (BaB) with specialized GPU-based bound propagation methods, and propose a *top-down beam-search* approach to quickly identify the subspace that may contain adversarial examples. Moreover, we build an *adversarial candidates pool* using cheap attacks to further assist the search in activation space via *diving* techniques and a *bottom-up* large neighbourhood search (LNS). Our adversarial attack framework, BaB-Attack, opens up a new opportunity for designing novel adversarial attacks not limited to searching the input space, and enables us to borrow techniques from integer programming theory and neural network verification to build stronger attacks. In experiments, we can successfully generate adversarial examples for hard input instances where existing strong adversarial attacks fail, and outperform off-the-shelf MIP solver based attacks in both success rates and efficiency. Our results further close the gap between the upper bound of robust accuracy obtained by attacks and the lower bound obtained by verification.

## 1 Introduction

Adversarial attacks aim to find adversarial examples (Szegedy et al., 2013), which are close to benign inputs in certain distance metrics yet trigger wrong behavior of neural networks (Carlini & Wagner, 2017; Madry et al., 2018; Athalye et al., 2018; Croce & Hein, 2020b). Adversarial attacks are important tools to gauge the empirical robustness of deep neural networks. Finding an adversarial example can be generally formulated as a constrained optimization problem:

$$x_{\text{adv}} = \arg \min_{x \in \mathcal{C}} f(x) \tag{1}$$

where $\mathcal{C}$ is often an $\ell_\infty$ or $\ell_2$ norm ball around the original input $x_0$, and $f(x)$ is an attack success criterion involving a neural network (such as the margin between the groundtruth class and another class): $f(x) < 0$ indicates a successful attack. A straightforward way of solving Eq. (1) is via first-order constrained optimization methods, such as projected gradient descent (PGD) (Madry et al., 2018) and its variants (Croce & Hein, 2020b; Tashiro et al., 2020; Croce & Hein, 2020a; Xie et al., 2019; Dong et al., 2018; Zheng et al., 2019). Additionally, some gradient-free attacks were proposed (Brendel et al., 2018; Cheng et al., 2018; Alzantot et al., 2019; Andriushchenko et al., 2020), mostly based on certain heuristic search on the input space $x$.

**Limitations of existing attacks.** As $f(x)$ usually consists of a highly non-convex neural network, solving Eq. (1) to its global minimum is challenging. This leads to failures in adversarial attacks: an adversarial example may exist but no attacks can find it, giving a false sense of security (Athalye et al., 2018). Especially, gradient based attacks can be easily trapped into a local minimum or misguided by masked gradients (Papernot et al., 2016; Tramèr et al., 2017). Even if we give the attacker *an infinite amount of time* (e.g., run a very large number of PGD steps, or allowing a large number of samples on input space), it is still hard to guarantee to find the adversarial example, since

it is *extremely difficult to systematically search* the high dimensional and continuous input space. Models concluded robust under existing attacks might still have security vulnerability in practice, leading to an urgent request for stronger attacks that can possibly approach ground-truth robustness.

**The mixed-integer approach.** This paper seeks stronger adversarial attacks from a different angle: instead of searching for adversarial examples in the input space, we look for adversarial examples in the *activation space*. The main intuition is that neural networks with piece-wise linear activation functions (e.g., ReLU) can be seen as a piece-wise linear function and each piece is uniquely defined by a specific setting of activation function status. For ReLU networks, each neuron can either be active (its input is positive) or inactive (its input is negative so the output is 0), which can be encoded by *discrete* 0-1 variables. This leads to a mixed integer programming (MIP) formulation (Ehlers, 2017), and Tjeng et al. (2019) conducted attacks using a MIP solver.

**Benefits of the MIP formulation.** The MIP formulation with the 0-1 encoding of ReLU neurons allows us to *systematically search* all the linear pieces in the input space, theoretically guarantee to *enumerate the entire input space* and obtain the global minimum of Eq. (1) given sufficient time. Practically, MIP-based attacks can often find adversarial examples that are missed by existing attacks and identify true weaknesses of a model. It also helps to *close the gap between the upper and lower bounds of robust accuracy* (i.e., attack accuracy vs. verified accuracy). Existing works aimed to reducing this gap by tightening the lower bound with stronger verifiers (Raghunathan et al., 2018b; Wang et al., 2021), while our work aims to tighten the upper bound by a systematic search of adversarial examples. Closing this gap is difficult even on small models (Dathathri et al., 2020).

**Generic MIP solvers are inefficient for adversarial attacks.** Despite its strengths, a MIP-based attack are often a few orders of magnitudes slower than existing attacks due to the high cost of running an off-the-shelf solver such as Gurobi (Tjeng et al., 2019). There are three root causes for its inefficiency. First, an off-the-shelf solver is *not aware of the underlying optimization problem corresponds to a neural network*, and has to apply generic solving techniques (e.g., using Simplex algorithm with relaxations) which can be expensive or ineffective. Second, it *cannot utilize solutions obtained cheaply from gradient based attacks* to accelerate its search. Third, generic MIP solvers are mostly restricted to CPUs and *can hardly utilize GPU acceleration*, which is crucial for efficiency.

**Contributions of this paper.** We address the above weaknesses in MIP solvers for adversarial attacks, by developing a GPU-accelerated branch and bound procedure to systematically search adversarial examples in the activation space, which is more efficient than generic MIP solvers. We focus on *solving hard instances*, where none of existing adversarial attacks based on searching the input space are successful but no verifier can prove their robustness. Our contributions includes:

• We apply the GPU-accelerated bound propagation based methods (Wang et al., 2021; Xu et al., 2020; Zhang et al., 2018; Wong & Kolter, 2018), which were originally developed for neural network verification, to the adversarial attack setting. These specialized methods can quickly *examine thousands of regions in activation space in parallel* and rule out the regions with no adversarial examples, which is difficult in off-the-shelf MIP solvers with a generic solving procedure.

• We employ a *top-down beam-search* to explore the activation space. Unlike the best-first search scheme used in many neural network verifiers, we can quickly go deep in the search tree and identify the most promising regions with adversarial examples. A smaller sub-MIP can then be created on-the-fly to search adversarial examples in the reduced regions with much fewer integer variables.

• We collect adversarial examples generated by cheap attacks in a *candidate pool* and utilize them in two ways. First, we conduct a *bottom-up* search on examples close to decision boundary by applying large neighborhood search (LNS). Second, when conducting the top-down search, we adopt *diving* by fixing integer variables according to adversarial examples in the pool to reduce the search space.

• Our new attack framework, BaB-Attack, is designed to tackle hard instances where existing strong adversarial attacks (such as AutoAttack) cannot succeed. Despite being more expensive than attacks on the input space, BaB-Attack is about an order of magnitude faster than using an MIP solver on hard instances, and can find adversarial examples that cannot be discovered by any existing attacks.

## 2 BACKGROUND

**Notations.** We define a $L$ layer feed-forward ReLU network as $f : \mathbb{R}^{n_0} \rightarrow \mathbb{R}$ and $f(x) := z^{(L)}(x)$, where $z^{(i)}(x) = \mathbf{W}^{(i)}\hat{z}^{(i-1)}(x) + \mathbf{b}^{(i)}$ with $i$-th layer weight matrix $\mathbf{W}^{(i)}$ and bias $\mathbf{b}^{(i)}$, $\hat{z}^{(i)}(x) = \text{ReLU}(z^{(i)}(x))$, and input $\hat{z}^{(0)}(x) = x$. Layer $i$ has dimension $n_i$, and $N$ is the total number of neurons. We denote the $j$-th neuron in layer $i$ as $z_j^{(i)}$. For a simpler presentation, we

assume $f(x)$ is a binary classifier and benign input $x_0$ has $f(x_0) > 0$. An attacker seeks to minimize $f(x)$ within a $\ell_\infty$ norm perturbation set $\mathcal{C}$ to make $f(x) < 0$. We can attack a multi-class classifier by considering each pair of target and ground-truth label individually where $f$ is defined as the margin between them, similarly to (Gowal et al., 2019b). We use $[N]$ to represent the set $\{1, \cdots, N\}$.

**The MIP formulation for adversarial attack.** Tjeng et al. (2019) formulated the adversarial attack and verification of ReLU network into a mixed integer programming (MIP) problem, solved by existing MIP solvers (refer to as the "MIP attack"). It has binary variables $s_j^{(i)}$ for each ReLU:

$$\min f(x) \text{ s.t. } z^{(i)}(x) = \mathbf{W}^{(i)}\hat{z}^{(i-1)}(x) + \mathbf{b}^{(i)}; \quad f(x) = z^{(L)}(x); \quad x \in \mathcal{C};$$
$$\hat{z}_j^{(i)}(x) \geq z_j^{(i)}(x); \quad \hat{z}_j^{(i)}(x) \leq u_j^{(i)} s_j^{(i)}; \quad \hat{z}_j^{(i)}(x) \leq z_j^{(i)}(x) - l_j^{(i)}(1 - s_j^{(i)}); \quad (2)$$
$$\hat{z}_j^{(i)}(x) \geq 0; \quad z_j^{(i)}(x) \in [l_j^{(i)}, u_j^{(i)}]; \quad s_j^{(i)} \in \{0, 1\}; \quad i \in [L], \, j \in [n_i]$$

where $s_j^{(i)}$ indicating the two status of a ReLU neuron: (1) *inactive*: when $s_j^{(i)} = 0$, constraints on $\hat{z}_j^{(i)}$ simplifies to $\hat{z}_j^{(i)} = 0$; or (2) *active*: when $s_j^{(i)} = 1$ we have $\hat{z}_j^{(i)} = z_j^{(i)}$. Here $l_j^{(i)}$, $u_j^{(i)}$ are pre-computed intermediate lower and upper bounds on pre-activation $z_j^{(i)}$ such that $l_j^{(i)} \leq z_j^{(i)}(x) \leq u_j^{(i)}$ for any $x \in \mathcal{C}$. The complexity of this problem can increase exponentially with the number of ReLU neurons, so it can take hours to run even on a small network, unless the network is trained with a strong regularization such as a certified dense (Wong & Kolter, 2018; Xiao et al., 2019).

**Searching in the Activation Space via Branch and Bound.**
Given the formulation in Eq. (2), we can view a neural network in the *activation space* $\mathcal{A} = [0, 1]^N$ where $N$ is the total number of neurons, and each dimension corresponds to the setting of a $s_j^{(i)} \in \{0, 1\}$ variable. To determine $s_j^{(i)}$ corresponding to a *known* adversarial example $x_{\text{adv}}$, we can propagate $x_{\text{adv}}$ through the network and check the sign of each neuron $z_j^{(i)}$, so $s_j^{(i)} = \mathbb{1}(z_j^{(i)} \geq 0)$. This uniquely locates the linear piece of $f(x)$ where $x_{\text{adv}}$ lies, because Eq. (2) becomes a set of linear inequalities when all $s_j^{(i)}$ are fixed. Intuitively, we can search adversarial examples by fixing all $s_j^{(i)}$ to one of the $2^N$ possible combinations in $\mathcal{A}$, and then solve Eq. (2) exactly using linear programming; an adversarial example is found when the solution is negative. To avoid clutter, we flatten the ordering of $s_j^{(i)}$ for $i \in [L], j \in [n_i]$ and use a single subscript $s_1, \cdots, s_N$ to denote all binary variables.

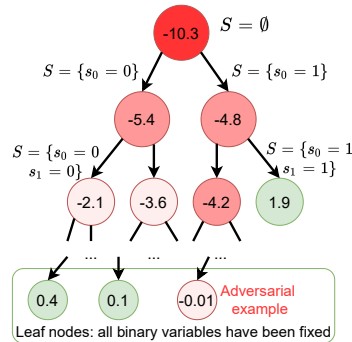

Figure 1: A branch and bound search tree. Each node represents a subdomain determined by $\mathcal{S}$, and the numbers are $\text{LB}(\mathcal{S})$. No adversarial example exist in the subdomain if $\text{LB}(\mathcal{S}) > 0$ (green). A concrete adversarial example is a leaf node $\mathcal{S}_{\text{leaf}}$ where $\text{LB}(\mathcal{S}_{\text{leaf}}) < 0$.

To effectively and systematically search in the activation space, instead of fixing all $s_i$ ($i \in [N]$), we can first fix a subset of them and bound the objective $f(x)$ to guide the search, leading to the *branch and bound* (BaB) method. In BaB, we solve Eq. (2) by creating subproblems constraining some binary variables, for example, $s_1 = 0$ or $s_1 = 1$ (since we can branch on the neurons in any fixed order, without loss of generality, we branch the neurons chronologically). We define a set $\mathcal{S}$ containing all the branching constraints (e.g., $\mathcal{S} = \{s_1 = 0, s_2 = 1\}$), which corresponds to a subdomain of the original problem Eq. (2). BaB requires the LB primitive on $\mathcal{S}$, which relaxes the remaining binary variables to obtain a lower bound of Eq. (2):

Lower bound in subdomain: $\text{LB}(\mathcal{S}) \leq \min f(x)$    s.t. $s \in \mathcal{S}$ and all other constraints in Eq. (2)

Here $s \in \mathcal{S}$ means setting binary variables $s_j^{(i)}$ according to the constraints in $\mathcal{S}$. Typically, more constraints in $\mathcal{S}$ lead to tighter bounds. $\text{LB}(\mathcal{S}) > 0$ indicates that *no adversarial example* exist within this subdomain, otherwise adversarial examples *may exist* in this subdomain.

We illustrate a BaB search tree in Fig. 1. Initially, $\mathcal{S}_{\text{root}} = \emptyset$, where $x \in C$ without any extra constraints in activation space and a lower bound of Eq. (2) is obtained. When $\text{LB}(\emptyset) < 0$, an adversarial example may exist, and we *branch* $\mathcal{S}_{\text{root}}$ into two subdomains:

$$\mathcal{S}_{1-} = \mathcal{S}_{\text{root}} \cup \{s_1 = 0\}; \; \mathcal{S}_{1+} = \mathcal{S}_{\text{root}} \cup \{s_1 = 1\}$$

Then we *bound* each subdomain. Since more constraints are added, $\text{LB}(\mathcal{S}_{1-})$ and $\text{LB}(\mathcal{S}_{1+})$ are usually improved. The branching procedure continues recursively, and if any $\text{LB}(\mathcal{S}) > 0$, no further

branching is needed since no adversarial examples are in that subdomain. Each branching increases the cardinality of $\mathcal{S}$ by 1, and eventually we reach leaf nodes with $|\mathcal{S}_{\text{leaf}}| = N$, each leaf locating a linear piece of $f(x)$. In that case, $\mathsf{LB}(\mathcal{S}_{\text{leaf}})$ is an exact solution since no binary variables are left, and if $\mathsf{LB}(\mathcal{S}_{\text{leaf}}) < 0$, a concrete adversarial example is the minimizer $x^*$ of Eq. (2) with $s \in \mathcal{S}_{\text{leaf}}$. Since $N$ can be quite large and adversarial examples lie in the leaf level, we must guide the search to reach there quickly. Although BaB is used in existing neural network verifiers (Bunel et al., 2018; De Palma et al., 2021a), they do not aim to reach the leaf level and typically branch the node with the worst bound first, generally leading to a wide but shallow search tree and unsuitable for detecting adversarial examples. We will discuss our search algorithm in Sec. 3.

**Bounding in Branch and Bound of Neural Networks.** The $\mathsf{LB}(\mathcal{S})$ primitive is crucial in the BaB process: it needs to provide a tight lower bound efficiently. A simple way to lower bound the objective of Eq. (2) is via relaxation of integer variables and linear programming (LP) (Bunel et al., 2018; Lu & Kumar, 2020); but an LP solver is needed which restricts its efficiency. Recently, a popular choice in neural network verifiers is the specialized bound propagation methods (Zhang et al., 2018; Wong & Kolter, 2018) which exploit the structure of the optimization problem (which a generic LP/MIP solver cannot) and give $\mathsf{LB}(\mathcal{S})$ efficiently on GPUs without an LP solver. Essentially, they relax each ReLU neuron into convex domains (Salman et al., 2019) and propagate them layer by layer through the network while maintaining sound bounds. A BaB and bound propagation based verifier $\alpha,\beta$-CROWN (Zhang et al., 2018; Xu et al., 2021; Wang et al., 2021), achieves the state-of-the-art verification performance (Bak et al., 2021), and we utilize its bounding subprocedure to produce $\mathsf{LB}(\mathcal{S})$. Importantly, we will show that how we use $\mathsf{LB}(\mathcal{S})$ to guide adversarial attacks, while existing works mostly use them for verification.

# 3 METHOD

**Overview of BaB attack.** To systematically search adversarial examples in activation space, we must explore the search tree and enumerate as more leaf nodes as possible. Although the worst case search time complexity is exponential in the numbers of ReLU neurons (visiting every leaf node of the tree), practically, if a right search procedure is chosen, only a small fraction of nodes need to be visited to find an adversarial example. In this paper, we propose BaB attack, specializing the BaB searching strategy over the activation space for the purpose of adversarial attacks. The search is well guided by (1) a top-down

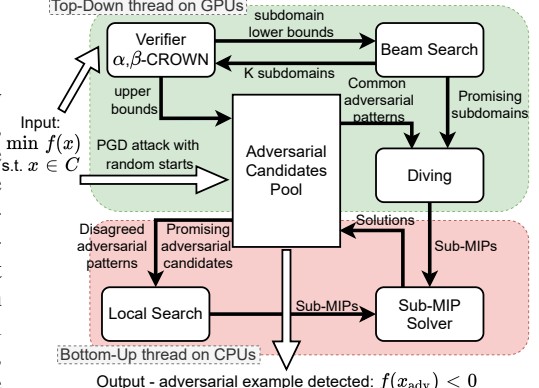

Figure 2: Overview of BaB attack.

beam-search thread accelerated on GPUs which quickly goes deep into the search tree, and (2) a bottom-up search thread on CPUs for large neighborhood search. The top-down and bottom-up searches run in parallel threads and they both benefit from the *adversarial candidates pool* $\mathcal{P}$, which contains examples $\mathcal{P} = \{x_{c_1}, \cdots, x_{c_M}\}$ where $M$ is the pool capacity. $f(x_{c_i})$ is still positive but small; the pool keeps the $M$ best (ranked by $f(x_{c_i})$; smaller is better) examples it receives. The activation space representations of these candidates are used as extra information to guide the search towards adversarial examples. We detail each part of BaB attack in next sub-sections.

## 3.1 TOP-DOWN BEAM SEARCH GUIDED BY NEURAL NETWORK VERIFIERS

**Challenges in searching adversarial examples.** The activation space can be quite large (e.g., with thousands or more dimensions), and adversarial examples are at the leaf level of the search tree. Searching directly from the root node and traversing the search tree in an exhaustive manner (such as BFS or DFS) can be quite insufficient. To locate adversarial examples faster, we propose to use beam search guided by lower bounds from neural network verifiers, as detailed below.

**Beam search in activation space.** The key insight in our procedure is to accelerate the search by prioritizing suspicious subdomains with small $\mathsf{LB}(\mathcal{S})$, quickly excluding the tree branches that are guaranteed with no adversarial examples. At the root node in the search tree, our beam search procedure expands the tree by $D$ levels, yielding $2^D$ subdomains:

$$\mathcal{S}_1 = \mathcal{S}_{\text{root}} \cup \{s_1 = 0, s_1 = 0 \cdots, s_D = 0\}, \quad \mathcal{S}_2 = \mathcal{S}_{\text{root}} \cup \{s_1 = 1, s_1 = 0 \cdots, s_D = 0\},$$
$$\mathcal{S}_3 = \mathcal{S}_{\text{root}} \cup \{s_1 = 0, s_1 = 1 \cdots, s_D = 0\}, \quad \cdots, \quad \mathcal{S}_{2^D} = \mathcal{S}_{\text{root}} \cup \{s_1 = 1, s_1 = 1 \cdots, s_D = 1\}$$

For each subdomain, we obtain its lower bound via the primitive $\text{LB}(\mathcal{S})$. We use the bound propagation procedure in the $\alpha,\beta$-CROWN verifier to efficiently provide $\text{LB}(\mathcal{S})$, but other efficient bounding methods can also be used in principle. Then, we sample $K$ subdomains without replacement out of the $2^D$ domains, with the following probability associated with each $\mathcal{S}_i$:

$$p_i = \frac{\exp\left(-T \cdot \text{LB}(\mathcal{S}_i)\right) \cdot \mathbb{1}(\text{LB}(\mathcal{S}_i) < 0)}{\sum_{i=1}^{2^D} \exp\left(-T \cdot \text{LB}(\mathcal{S}_i)\right) \cdot \mathbb{1}(\text{LB}(\mathcal{S}_i) < 0)}, \; i \in \{1, \cdots, 2^D\}, \; T \text{ is the temperature, set to 0.1}$$

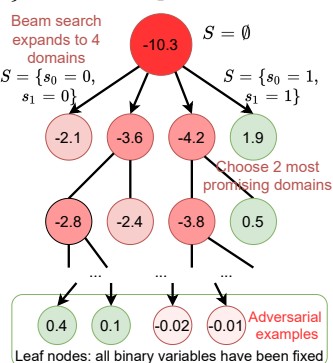

A subdomain with more negative lower bound has a higher probability to be selected, since the large negative bounds may indicate a higher chance of the existence of adversarial examples. Subdomains with positive bounds will never be selected, since they are guaranteed to not contain adversarial examples. The picked out subdomains $\mathcal{S}'_1, \cdots, \mathcal{S}'_K$ become the parent nodes for the next iteration of beam search. In the next iterations, we explore $K \cdot 2^D$ subdomains and increase the depth by $D$ per iteration. Since all the subdomain lower bounds are computed in a large batch on GPUs and only take a few seconds, our search procedure quickly explore deep in the search tree. Our *specialized top-down beam search* procedure with lower bounds computed efficiently by *bound propagation methods on GPUs* brings us great advantages over existing MIP solvers, which conduct BaB on CPUs using a generic procedure such as the Simplex algorithm. In practice, we can visit several orders of magnitude more subdomains.

Figure 3: Beam search: we select $K$ subdomains probabilistically according to $\text{LB}(\mathcal{S})$, and expand the search tree by $D$ levels using bound propagation on GPU.

**Sub-MIP on most promising subdomains.** Before the beam search reaches the leaf level, we start searching adversarial examples in the most promising subdomains (e.g., in some of the selected domains $\mathcal{S}'_1, \cdots, \mathcal{S}'_K$), by constructing a sub-MIP problem of Eq. (2): in the $k$-th sub-MIP, we fix its binary variables $s_j^{(i)}$ based on constraints in $\mathcal{S}'_k$. In this step, although a generic MIP solver is used, it is instructed to search in subdomains guided by beam search to be likely to contain adversarial examples. With a large number of $s_j^{(i)}$ fixed during beam search, the MIP solver only needs to work on a much smaller problem and can be much more effective than solving Eq. (2) directly.

**Completeness.** Beam search, if implemented with back-tracking, can achieve completeness (Zhou & Hansen, 2005): given sufficient time, it will systematically visit all leaf nodes, and guarantee to either locate an adversarial example, or prove the network safe. However, due to the large number of neurons and associated binary variables, achieving completeness often requires an infeasibly large amount of time and space. We thus focus on searching adversarial examples as fast as possible rather than exhaustively visiting every node, although theoretically our procedure can be made complete.

## 3.2 Diving in Branch and Bound with Common Adversarial Patterns

**What is diving?** "Diving" refers to diving deep in the BaB search tree by heuristically fixing some integer variables without exploring all possible branches. It is a common strategy in generic MIP solvers, able to quickly uncover feasible solutions of a MIP problem (Berthold, 2006; Nair et al., 2020). In our case, we want to fix binary variables $s_j^{(i)}$ in Eq. (2), and a feasible solution $x$ with $f(x) < 0$ is an adversarial example (Figure 4). A generic MIP solver uses diving to hopefully find high quality feasible solutions quickly, however it *cannot use the information provided by cheaply generated adversarial examples* like PGD attack to guide this heuristic. In this work, we propose a specialized diving scheme in the activation space based on the statistics in the adversarial candidates pool, and construct sub-MIPs with additional diving constraints to reduce the search space.

**Diving with common adversarial patterns.** Given the candidates pool $\mathcal{P} = \{x_{c_1}, \cdots, x_{c_M}\}$, we first extract the corresponding binary variables $s_i$ for each example, by propagating them through the network (see Section 2). The binary variable corresponds to the $i$-th neuron of the $m$-th adversarial example is denoted as $s_{i,m}$ (0 or 1). A variable $s_i$ is called a *common activation* when the function $c(i)$ is greater than a threshold:

$$c(i) := \frac{\left|\sum_{m=1}^{M} s_{i,m} - M/2\right|}{M} + 0.5 \geq C, \quad C \in [0.5, 1.0]$$

For example, when $C = 0.9$, a common activation requires that at least 90% examples in the pool share the same value of $s_i$ (0 or 1). All the common activations and their common values (0 or 1)

are called *common adversarial patterns*, indicating that an adversarial examples will be very likely to contain these settings of $s_i$. We can thus construct a set of constraints $\mathcal{S}_{\text{common}}$, setting common activations and their common values:

$$\mathcal{S}_{\text{common}} = \{s_i = \mathbb{1}(\sum_{m=1}^{M} s_{i,m} \geq \frac{M}{2}), c(i) \geq C, i \in [N]\}$$

Then, when constructing the top-down sub-MIPs in Sec. 3.1, we provide additional constraints $\mathcal{S}_{\text{common}}$. Including these additional constraints further reduces the search space for the MIP solver (the MIP solver solves Eq. (2) with both beam search constraints $\mathcal{S}'_1$ and diving constraints $\mathcal{S}_{\text{common}}$), making it easier to find an solution. The threshold $C$ controls the aggressiveness of diving; too much diving may lead to a too small search space so good adversarial examples cannot be found.

**How to fill the adversarial candidates pool?** To obtain useful common adversarial pattern, we maintain an *adversarial candidates pool* with up to $M$ most promising adversarial candidates ($f(x)$ is close to 0 but the label is not yet flipped). The pool is initialized with perturbed samples bounded within $\mathcal{C}$ found with output diversified PGD (Tashiro et al., 2020), which creates a diverse set of adversarial candidates. During our BaB attack process, new adversarial candidates come from three sources: (1) some neural network verifier provides upper bounds for each subdomain during beam search; in $\alpha,\beta$-CROWN these upper bounds are obtained via conducting PGD on its dual solutions (Wang et al., 2021) produced by the solver, and they are added to the pool; (2) the solutions returned by the top-down sub-MIP solved with beam search and diving constraints, and (3) the solutions returned by the bottom-up sub-MIP with large neighborhood search (which will be discussed in the next subsection). When new adversarial candidates are inserted, they are compared to existing ones in the pool, and we keep the best $M$ adversarial candidates with distinct activation patterns.

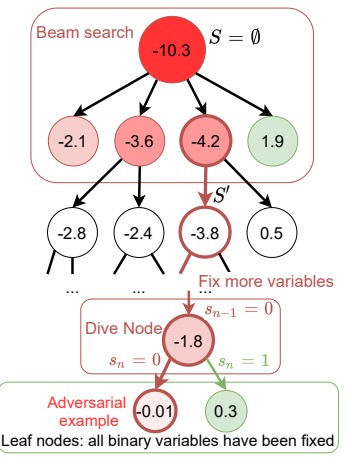

Figure 4: We dive the search tree with additional constraints constructed by common adversarial patterns to greatly reduce top-down sub-MIP search space.

### 3.3 BOTTOM-UP LARGE NEIGHBORHOOD SEARCH (LNS)

**What is bottom-up search?** The bottom-up search procedure starts at the leaf nodes of the BaB search tree (Figure 5): we start from a known adversarial candidate $x_c$ that is close to decision boundry ($f(x_c) > 0$ but very small), and want to further reduce $f(x_c)$ by searching around $x_c$. A naive way is to conduct PGD attack in the input space with $x_c$ as the starting point, but we found it not helpful because the adversarial candidates in the pool have already been optimized using PGD or stronger attacks. Thus, we propose to use a large neighborhood search *in the activation space*.

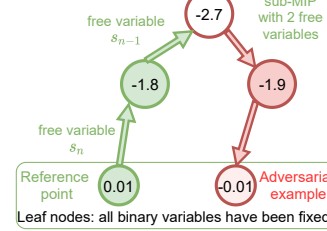

Figure 5: In bottom-up search, we *free* some fixed integer variables at a leaf node (an adversarial candidate) searching its neighborhood in activation space.

**Bottom-up search via large neighborhood search.** Large neighborhood Search (LNS) (Walser, 2003; Schrijver, 2003) is a generic local search heuristic: one defines a neighborhood around a reference point (a feasible solution) and finds the optimum objective in this neighborhood, typically by constructing a sub-MIP problem with neighborhood constraints. In the setting of integer programming, the neighborhood can be defined by *freeing* certain fixed integer variables, allowing them to be optimized while fixing other integer variables. However, traditional local search algorithms in MIP solvers has little guidance regarding promising subdomains in common adversarial examples and can be ineffective due to the high dimensional search space in the adversarial attack problem.

In BaB attack, we extend the general idea of LNS to a specialized local search for adversarial attacks by selecting the most promising adversarial candidates in the pool as the reference point and then use the statistics from the pool to free certain binary variables $s_i$. Specifically, among all the binary variables corresponding to ReLU neurons on the selected candidate, we find the ReLU neurons where adversarial candidates in the pool that *disagree the most* and define the *disagreed adversarial patterns*. Formally, similar to the setting in Sec. 3.2, a variable $s_i$ is called a *disagreed activation* if:

$$\bar{c}(i) := 1 - c(i) \geq \bar{C}, \quad \bar{C} \in [0.0, 0.5]$$

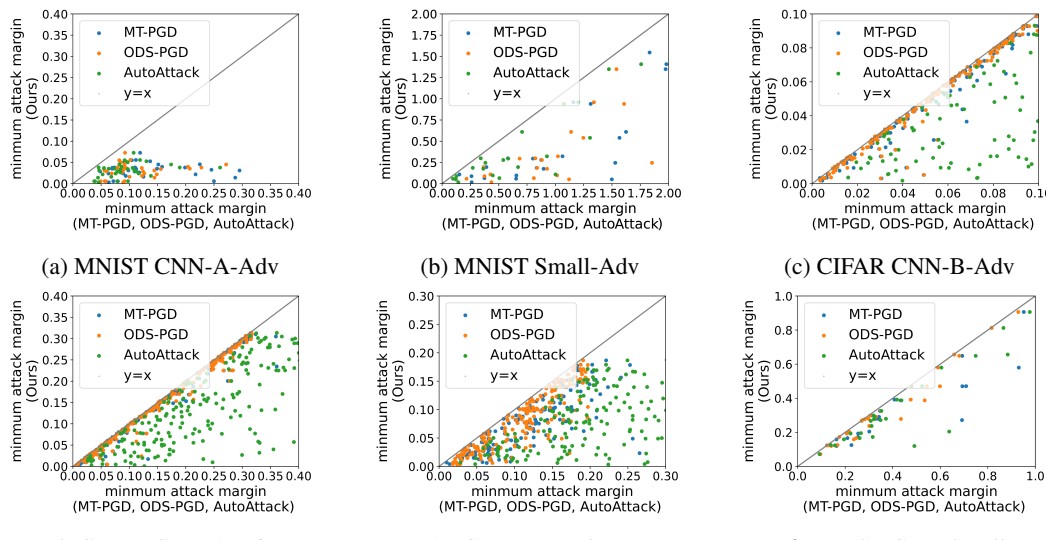

Figure 6: For examples that all attacks failed, we compare the minimum margin between the ground-truth label and all other target labels for the adversarial candidate. A smaller margin is better. Our attack achieves noticeably smaller margins compared to other attacks (margins from other attacks are below the $y = x$ line).

For example, when $\bar{C} = 0.3$, $s_i$ is a disagreed activation when there are at least 30% examples in the pool do *not* share the same value of $s_i$. Since these variables are quite different across existing adversarial examples, we remove their corresponding binary variables to allow the MIP solver to search for a better setting of these variables. The aggressiveness of freeing variables in LNS is determined by $\bar{C}$. The set of disagreed adversarial patterns is a set of binary variables:

$$S_{\text{disagreed}} = \{s_i \mid \bar{c}(i) \geq \bar{C}, i \in [N]\}$$

Formally, to search around an adversarial candidate $x_c$, we first propagate $x_c$ through the network and obtain activation values $z_i$, extract the corresponding binary variables $s_i$ for $x_c$, and remove the constraints that are in $S_{\text{disagreed}}$ to construct the set of constraints for bottom-up search:

$$\mathcal{S}_{\text{bottom-up}} = \{s_i = \mathbb{1}(z_i \geq 0), s_i \notin S_{\text{disagreed}}, \ i \in [N]\}$$

We then construct a sub-MIP using Eq. (2) with the additional constraints $\mathcal{S}_{\text{bottom-up}}$ and solve it using a MIP solver. The optimal solution to each sub-MIP (if still not an adversarial example) will be added back to adversarial candidates pool again waiting for another round of local search.

## 4 EXPERIMENTS

**Setup.** We evaluate our methods on all 10,000 test examples of MNIST (LeCun, 1998) and CIFAR10 (Krizhevsky et al., 2009) dataset, and select 9 models which are mostly benchmarking models used in previous works or competitions (details in Appendix A). For each model, we first run three commonly used strong adversarial attacks: a multi-targeted PGD (MT-PGD) attack (Gowal et al., 2019a) with 100 Adam steps and 30 random restarts; a multi-targeted PGD attack with Output Diversified Sampling (ODS-PGD) (Tashiro et al., 2020) with 100 Adam steps and 30 random restarts, and AutoAttack (Croce & Hein, 2020b) which is an *ensemble* of parameter-free attacks. The remaining robust images are then tested with $\alpha, \beta$-CROWN verifier (Wang et al., 2021; Xu et al., 2021; Zhang et al., 2018) to see if they can be verified robust so no further attack is needed. Any images that failed with the verifier are then evaluated in an MIP formulation (Tjeng et al., 2019) solved using Gurobi. The MIP solver is used as the last resort because it is usually much more expensive than other approaches. We set an one hour timeout for MIP attack and our BaB attack, but our attack usually terminates much faster than a generic MIP solver. Any instance that cannot be attacked via a combination of MT-PGD, ODS-PGD and AutoAttack nor verified is referred to as a *hard instance*, and the main evaluation is conducted on these instances. All attacks are $\ell_\infty$ norm-based with $\epsilon$ listed in Table 1. The hyperparameters can be found in Appendix A.2.

**Results.** Table 1 shows a breakdown of all 10,000 test examples on the 5 models. Among these models, MNIST Small-Adv, MNIST Conv-Small, CIFAR $LP_D$-CNN-A, and $LP_D$-ResNet are relatively easy, and most of datapoints can be either attacked or verified by SOTA tools. The $LP_D$-CNN-A and $LP_D$-ResNet models were among the largest ones in (Tjeng et al., 2019), however they have become quite easy due to the recent progress of neural network verifiers - only 2 and 1 (out of 10,000) examples are hard instances. On the relatively small MNIST Small-Adv model, a MIP solver can terminate within a reasonable amount of time, and we can attack the same number of images while being faster. On the MNIST CNN-A-Adv model, many instances timeout with the MIP solver and we can attack more images under less time; see Fig. 7 and Fig. 8 (in Appendix) for time vs. number of images attacked. The CIFAR CNN-B-Adv, CNN-A-Mix, and Marabou models are much harder, and we managed to solve 5, 17, and 27 more images compared to MIP attack within much shorter time. BaB attack can solve the same number of images as MIP attack on MNIST Conv-Small and CIFAR CNN-A-Adv but is over 10x faster in average. For images that none of the attacks work, we plot the minimum margin between the ground-truth label and other labels in Fig. 6 and compare the margins against ODS-PGD, MT-PGD attacks and AutoAttack. Our method consistently achieves smaller margins on all 6 models.

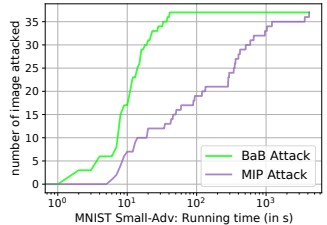

(a) MNIST Small-Adv

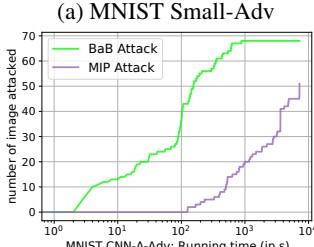

(b) MNIST CNN-A-ADV

Figure 7: Running time vs. number of attacked images compared to MIP attack.

**Comparison to more attacks.** We compare our BaB attack to 5 state-of-the-art attacks on all hard instances in Table 2. Attacks we evaluated include 1) FAB attack (Croce & Hein, 2020a): a white-box boundary adversarial attack; 2) Square attack (Andriushchenko et al., 2020): a query-efficient black-box adversarial attack using randomized search with localized square-shaped updates; 3) DIFGSM attack (Xie et al., 2019): a white box attack leveraging input diversity; 4) MIFGSM attack (Dong et al., 2018): a momentum-based iterative white box attack to escape from poor local maxima and 5) Distributional attack (Zheng et al., 2019): a white box attack considering adversarial-data distribution. We increased steps and iterations to make some attacks stronger; hyperparameters are in Appendix A.3. Most attacks are not effective on these hard instances. Square attack is the strongest among them but still finds less adversarial examples than us and is also relatively slow.

**Ablation study.** To fully understand how each proposed component contributes to the overall performance of BaB attack, we conduct ablation study on MNIST Small-Adv and MNIST CNN-A-Adv models and show the number of successfully attacked instances and average time in Table 3. We first show that the beam search guided by verifier $\alpha,\beta$-CROWN significantly helps the performance compared to random top-down search without any guidance. We then show that the top-down diving and bottom-up large neighborhood search contribute to different hard instances and the combination of them lead to the best attack performance. Note that the average time here is on successfully attacked samples by each method only, showing the efficiency of each component instead of strength.

| Dataset | Model | eps | Clean Acc. | Total verified | Total attacked | Hard instances | MIP attack | Avg. time(s) | BaB attack | Avg. time(s) |
|---|---|---|---|---|---|---|---|---|---|---|
| MNIST | Small-Adv | 0.3 | 97.94% | 8176 | 1559 | 59 | **37** | 439.88 | **37** | 16.15 |
| | CNN-A-Adv | 0.3 | 96.33% | 6757 | 2239 | 637 | 51 | 2546.13 | **67** | 111.77 |
| | Conv-Small | 0.12 | 98.04% | 7628 | 2159 | 38 | 17 | 3358.49 | 17 | 8.52 |
| CIFAR | LPd-CNN-A | 2/255 | 60.86% | 5019 | 1065 | 2 | **2** | 13.80 | **2** | 9.90 |
| | LPd-RES | 8/255 | 27.07% | 2243 | 463 | 1 | 0 | - | 0 | - |
| | CNN-A-Adv | 2/255 | 65.63% | 4755 | 1660 | 148 | 8 | 906.77 | **8** | 57.98 |
| | CNN-B-Adv | 2/255 | 78.25% | 4539 | 1735 | 1551 | 1 | 3661.40 | **6** | 65.61 |
| | CNN-A-Mix | 2/255 | 74.18% | 4298 | 2058 | 1062 | 2 | 3467.52 | **19** | 71.65 |
| | Marabou | 2/255 | 63.14% | 875 | 4188 | 1251 | 15 | 1470.22 | **42** | 56.82 |

Table 1: Comparison between MIP attack (Tjeng et al., 2019) (using Gurobi) and our BaB Attack. Both attacks focus on *hard instances* where a combination of MT-PGD, ODS-PGD and AutoAttack cannot attack and their robustness also cannot be verified using a verifier. We are faster than MIP Attack and can also find more adversarial examples in all models. Average time excludes examples where all methods time out.

## 5 RELATED WORK

**Adversarial examples and attacks.** Adversarial examples were first discovered in (Szegedy et al., 2013; Biggio et al., 2013) and they can be easily constructed by single-/multi-step gradient descent to fool regularly trained neural networks (Kurakin et al., 2016; Goodfellow et al., 2015). However, purely gradient-based methods can fail due to gradient masking and obfuscated gradients (Tramèr et al., 2017; Papernot et al., 2016; Athalye et al., 2018). Popular attacks like PGD (Madry et al., 2018) or CW (Carlini & Wagner, 2017) can lead to overestimation of robustness (Mosbach et al.,

| Dataset | Model | # Total | FAB | | Square | | DIFGSM | | MIFGSM | | Distributional | | BaB attack | |
|---|---|---|---|---|---|---|---|---|---|---|---|---|---|---|
| | | | # succ. | Time(s) | # succ. | Time(s) | # succ. | Time(s) | # succ. | Time(s) | # succ. | Time(s) | # succ. | Time(s) |
| MNIST | Small-Adv | 59 | 1 | 32.29 | 12 | 79.61 | 0 | 0.04 | 0 | 0.02 | 0 | 0.03 | **37** | 16.15 |
| | CNN-A-Adv | 637 | 0 | 33.96 | 17 | 117.51 | 0 | 0.04 | 0 | 0.02 | 1 | 0.01 | **67** | 111.77 |
| | Conv-Small | 38 | 1 | 31.90 | 14 | 113.03 | 0 | 0.04 | 0 | 0.02 | 1 | 0.03 | **17** | 8.52 |
| CIFAR | LPd-CNN-A | 2 | 0 | 34.61 | 0 | 117.33 | 0 | 0.02 | 0 | 0.02 | 0 | 0.64 | **2** | 9.90 |
| | LPd-RES | 1 | 0 | 48.98 | 0 | 242.52 | 0 | 0.42 | 0 | 0.06 | 0 | 1.62 | 0 | - |
| | CNN-A-Adv | 148 | 0 | 33.22 | 0 | 116.76 | 0 | 0.39 | 0 | 0.02 | 0 | 0.01 | **8** | 57.98 |
| | CNN-B-Adv | 1551 | 3 | 16.81 | 3 | 119.88 | 2 | 0.01 | 2 | 0.02 | 2 | 0.01 | **6** | 65.61 |
| | CNN-A-Mix | 1062 | 0 | 34.13 | 0 | 117.16 | 0 | 0.04 | 0 | 0.02 | 0 | 0.02 | **19** | 71.65 |
| | Marabou | 1251 | 0 | 29.21 | 0 | 111.70 | 0 | 0.04 | 0 | 0.02 | 1 | 0.03 | **42** | 56.82 |

Table 2: Number of successfully (# succ.) attacked *hard instances* (from Table 1) under five more attacks.

| | MNIST-Small-Adv | | | MNIST-CNN-A-Adv | | |
|---|---|---|---|---|---|---|
| | # success | # total | Avg. time | # success | # total | Avg. time |
| Random top-down beam search | 8 | 59 | 19.52 | 22 | 637 | 13.46 |
| Verifier guided beam search | 36 | 59 | 21.58 | 54 | 637 | 203.75 |
| Verifier guided beam search + diving | 36 | 59 | 20.63 | 57 | 637 | 105.49 |
| Verifier guided beam search + bottomup search | 36 | 59 | 22.67 | 66 | 637 | 110.65 |
| Verifier guided beam search + diving + bottomup search | 37 | 59 | 16.15 | 67 | 637 | 111.77 |

Table 3: Ablation study on how each proposed component contributes to the overall performance.

2018; Croce et al., 2020) despite their empirically good performance and efficiency. A large body of white-box adversarial attacks have been proposed to strengthen adversarial attacks; many of them are variants of PGD based attacks (Zheng et al., 2019; Tashiro et al., 2020; Gowal et al., 2019b; Wang et al., 2019). Due to non-convexity of the adversarial attack objective, gradient-free methods and black-box attacks are also widely explored but mostly result in similar or worse performance compared to gradient-based ones (Papernot et al., 2017; Chen et al., 2017; Ilyas et al., 2018a;b; Xiao et al., 2018; Andriushchenko et al., 2020). Recently, stronger attacks (Croce & Hein, 2020a;b) are proposed but they are also restricted to searching in input space. In this paper, we are the first to conduct a systematic and efficient search of adversarial examples in activation space inspired by branch and bound techniques used for integer optimization and neural network verification.

**Neural network verification.** Early neural network verifiers solve the verification problem with satisfiability modulo theories (SMT) or MIP solvers and can only scale to very small networks (Katz et al., 2017; Huang et al., 2017; Ehlers, 2017; Dutta et al., 2018; Tjeng et al., 2019). Efficient verification methods with various sound relaxations are then proposed for verifying larger networks but without completeness guarantee (Wong & Kolter, 2018; Dvijotham et al., 2018; Raghunathan et al., 2018a;b; Singh et al., 2018b;a; Zhang et al., 2018; Tjandraatmadja et al., 2020). Branch and bound (BaB) based verifiers can efficiently branch on ReLU neurons and achieve completeness on ReLU networks using efficient incomplete verifiers (Bunel et al., 2018; Wang et al., 2018a; Lu & Kumar, 2020; Botoeva et al., 2020). BaB based methods with input domain split and refinements are also investigated but they are limited to low input dimensions (Wang et al., 2018b; Royo et al., 2019; Anderson et al., 2019). Recent verifiers use GPU accelerated incomplete solvers to further scale up verification and achieve several orders of magnitude speedup (Bunel et al., 2020; De Palma et al., 2021b; Anderson et al., 2020; Xu et al., 2021; Müller et al., 2021; 2020; Wang et al., 2021). Our BaB attack relies on the lower bounds of BaB subdomains computed by $\alpha,\beta$-CROWN (Zhang et al., 2018; Xu et al., 2021; Wang et al., 2021), one of the most efficient GPU-based verifiers that won the second neural network verification competition (VNN-COMP 2021) (Bak et al., 2021).

## 6 CONCLUSION

In this paper, we proposed BaB attack, a strong adversarial attack using branch and bound to systematically search adversarial examples in activation space. We propose a top-down beam search utilizing bound propagation methods from neural network verifiers on GPUs to quickly locate potential subdomains containing adversarial examples. We further exploit the common activation patterns of adversarial candidates obtained by cheap attacks to guide the searching process with diving. Moreover, a bottom-up large neighborhood search procedure explores around promising adversarial candidates to find potential adversarial examples. Experimental results show that our BaB attack can find adversarial examples for hard input instances where SOTA adversarial attacks fail. We achieve higher attack success rates within much less time compared to generic MIP solver based attacks.

**Limitations of this study.** One limitation of BaB Attack is speed - it usually takes a few minutes to explore a branch and bound tree using beam search, while gradient based adversarial attacks are often very fast, in a few seconds. Practically, BaB attack is mostly useful for hard instances, and we can use other simpler attacks as a filter to reduce the number of inputs for BaB attack. Additionally, BaB attack currently relies on neural network verifiers to give a lower bound for each subdomain to guide the beam search, so its scalability is limited by SOTA verification tools. Yet, we have demonstrated that it is faster and also more effective than attacks using an off-the-shelf MIP solver.

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

# A APPENDIX

## A.1 DETAILS OF MODELS

MNIST CNN-A-Adv, CIFAR CNN-A-Adv, CIFAR CNN-B-Adv and CIFAR CNN-A-Mix models are provided by (Dathathri et al., 2020) and were trained with adversarial training, except that CIFAR CNN-A-Mix is trained with a mixture of adversarial training loss and certified defense loss. These models were used as a benchmark to evaluate the gap between verified accuracy and attack acucracy in a few papers (Dathathri et al., 2020; Wang et al., 2021; Müller et al., 2021). MNIST Small-Adv is trained using adversarial training with architecture similar to the MNIST model used in (Madry et al., 2018) but scaled down by 4X and maxpool layers removed. MNIST Conv-Small is a naturally trained model from the ERAN benchmarks (Singh et al., 2018a; Müller et al., 2021). CIFAR LPd-CNN-A and LPd-RES (Tjeng et al., 2019) are trained using a dual linear programming based certified defense (Wong et al., 2018), so they are relatively easy to attack and verify. The CIFAR Marabou model is from the `marabou-cifar10` benchmark in 2nd International Verification of Neural Networks Competition (VNN-Comp) 2021 (Bak et al., 2021); the model (original name `cifar10_small.onnx`) is naturally trained on CIFAR-10 dataset.

## A.2 HYPERPARAMETERS FOR OUR BAB ATTACK

We set the common adversarial pattern threshold $C$ to be 1.0 for both top-down diving and disagreed adversarial pattern threshold $\tilde{C}$ to 0.0 for bottom-up large neighborhood search. We use state-of-the-art verifier $\alpha,\beta$-CROWN to provide verified lower bounds $\text{LB}(\mathcal{S})$, prioritizing suspicious subdomains. We use the default learning rates 0.01 and 0.05 for both $\alpha$-CROWN and $\beta$-CROWN. To tighten the estimation for each subdomain and provide more accurate guidance for suspicious ones, we increase the optimization iterations to 100 for both $\alpha$-CROWN and $\beta$-CROWN with a learning rate decay of 0.999. We use the maximal batch size $B$ to fit into the GPU memory. For each step of our beam search, we set the depth for each iteration of beam search $D$ to be 8 and the number of picked out subdomains $K$ to be $B/2^D$. One can adjust $D$ to control the searching speed.

We run all our experiments on a EPYC 7502 CPU with 32 physical cores, and use up to 32 sub-MIP threads for top-down or bottom-up search. We set a timeout threshold for each sub-MIP to be 30 seconds for MNIST-Small-Adv (because it is a relatively small model), 180s for the other MNIST models and 360s for CIFAR10 models. We run each attack for up to an hour to be consistent with the timeout threshold of baseline MIP attack while our BaB attack usually terminates much faster.

## A.3 SETUP FOR COMPARED ATTACKS

We did a small scale grid search for the best combination of hyperparameters for each attack. For FAB attack, we select a 100-steps attack with 50 restarts and set $\alpha_{max} = 0.1$, $\beta = 0.9$ and $\eta = 1.05$; for Square attack, we use 7000 queries, 20 restarts and set 0.6 as the size of squares. Note that FAB and Square attacks are part of the AutoAttack suite, but here we significantly increased their steps and number of iterations compared to the defaults in AutoAttack. For DIFGSM attack, we set $\alpha = 6/255$, decay rate as 0.9, number of iterations as 50 and the probability of applying input diversity as 0.5; for MIFGSM attack, we set $\alpha = 2/255$, decay rate as 0.9 and number of iterations as 20; for Distributional attack, we used 100 attack iterations, step size 0.01, and a balancing parameter of 0.05.

## A.4 MORE EXPERIMENTAL RESULTS

We plot running time vs. number of attacked images for five more models in Figure 7. Our BaB attack achieves distinctly faster running time and produces more successful attacked images compared to the MIP attack using an off-the-shelf sovler.

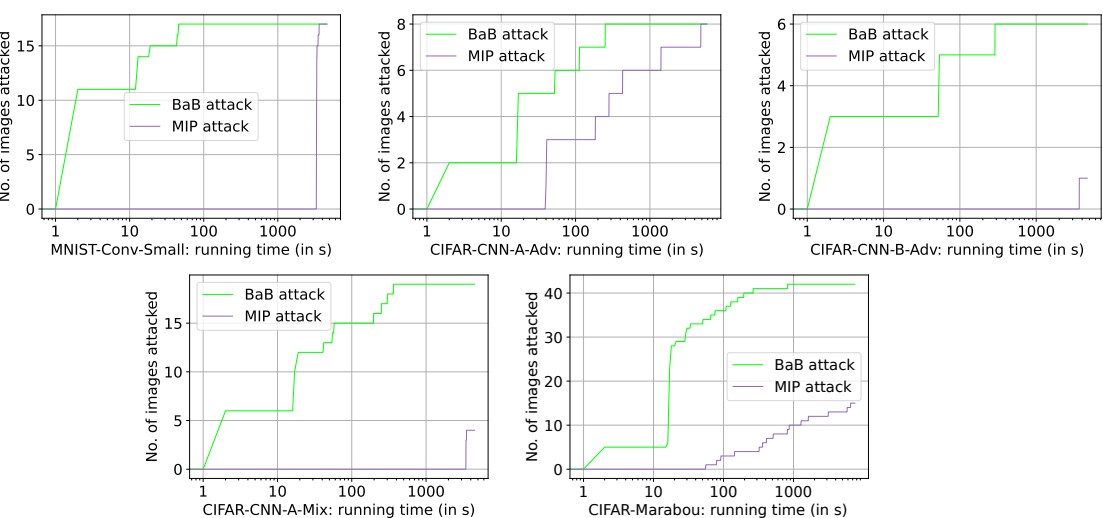

Figure 8: Running time vs. number of attacked images compared to MIP attack on five models.

