# OpenReview forum: "A Branch and Bound Framework for Stronger Adversarial Attacks of ReLU Networks"
_ICLR.cc/2022/Conference — ICLR 2022 Submitted_

### Official Review · Reviewer_aJSd · 2021-11-02

**Correctness:** 4
**Technical Novelty And Significance:** 3
**Empirical Novelty And Significance:** 3
**Recommendation:** 6
**Confidence:** 2

**Details Of Ethics Concerns:**

New adversarial attack is presented with all corresponding concerns.

**Main Review:**

**Strengths:**
1) New interesting way to approach adversarial attacks from a mixed integer programming perspective (although not fully clear why).
2) Presented several heuristics that improves practicality and speed of the adversarial attack search.

**Weaknesses:**
1) Very difficult to understand the problem that the paper is trying to solve, the main idea, why the results are meaningful.
Why integer programming and such an esoteric formulation for adversarial attack search?
What are the benefits of this formulation and method compared to the standard gradient based approach?
I suggest rewriting the paper, especially the background part, to make it more understandable and accessible.
2) Minor grammatical and punctual errors - “**an binary** encoding”, “locate adversarial examples **if exist**”, “**a**dversarial candidates pool dynamically filled...”, etc.


**Summary Of The Paper:**

Authors proposed a novel branch and bound attack, which searches adversarial examples in the activation space of binary variables in a mixed integer programming formulation. They also present several heuristics such as top-down beam-search, diving, and bottom-up large neighborhood search guided by the adversarial candidates pool to efficiently obtain strong adversarial attacks better than standard MIP solvers and other SOTA methods.

**Summary Of The Review:**

Since it is hard to capture the main point of the paper and the problem it is trying to solve, I tend to be on the rejection side.

---

> ### Author Response · Authors · 2021-11-22
> **We have rewrote part of the paper: the motivation is much more clear and the paper is more accessible**
>
> Thank you for your helpful review and your comments helped us a lot in improving our paper. We have **rewritten part of the paper to make it more accessible** (especially the introduction and background) and we hope you can take a look at our revision. Let us also summarize the main motivation of our paper here:
>
> ### Motivation: why is integer programming and such an esoteric formulation useful? What is the benefits of this formulation compared to the standard gradient based approach?
>
> Existing attacks search in the input space (e.g., using gradient descent). Due to non-convexity, it is **hard to systematically or exhaustively search a continuous and high dimensional input space**, so even given an attack a very long time (e.g., run PGD attack for 10,000 iterations or 10,000 random start points), it can still miss some adversarial examples.
>
> We formulate the attack in the activation space, by encoding the ReLU neurons as 0-1 variables in a mixed integer programming (MIP) formulation. The problem is discrete so we can **systematically search it via branch and bound and find more adversarial examples**. MIP based attacks are often stronger than other attacks. Given an infinite amount of time, it can guarantee to find an adversarial example if one exists thanks to the systematic search procedure.
>
> We hope the reviewer can kindly read our updated Introduction section for a more detailed discussion.
>
> ### What is the major contribution of our work?
>
> When solving the activation space attack using a generic MIP solver like Gurobi or CPLEX, the **generic MIP solver can be very inefficient** because it cannot exploit the structure of the neural networks to speed up solving,  cannot utilize information from cheap adversarial attacks (e.g., PGD attacks), and also can hardly be accelerated on GPUs.  So although attacks on activation space can be powerful, they are extremely slow using a generic MIP solver.
>
> **Our work solves these weaknesses** using a specialized top-down beam search and bottom-up large neighborhood search. We use bound propagation methods to exploit the structure of the networks to speedup solving; we maintain a pool of existing adversarial examples to guide the search in activation space; our search is also accelerated on GPUs via specialized neural network verifiers.
>
> By overcoming these drawbacks in a generic MIP solver, we greatly improved adversarial example search on the activation space: we can **find more adversarial examples and also be over a magnitude faster** than the generic MIP solver on most models.
>
> We hope the reviewer can understand our motivation and contribution better, and please kindly let us know if any part is still unclear to you.
>
> Finally, we have also fixed many minor typos including those found by you (thanks!). Additionally, as suggested by other reviewers, we’ve **also added significantly more experiments**. We hope the reviewer can **check out our revised paper (especially the introduction)** and reevaluate based on the new revision. Please feel free to ask us if you have any additional questions or anything is still unclear.

---

> > ### Comment · Reviewer_aJSd · 2021-11-29
> > **Thank you for addressing my concerns!**
> >
> > I believe the authors addressed my main concern about the motivation of their method, thus I increased my score.

---

> > > ### Author Response · Authors · 2021-11-29
> > > **Thank you for your response!**
> > >
> > > Dear Reviewer aJSd,
> > >
> > > We greatly appreciate your encouragement and we are glad to know that your concerns are addressed. Thank you for positively supporting our paper during the discussion period and please kindly let us know if there are any additional questions.
> > >
> > > Sincerely,
> > > Paper 2303 Authors

---

### Official Review · Reviewer_LiPk · 2021-11-02

**Correctness:** 4
**Technical Novelty And Significance:** 3
**Empirical Novelty And Significance:** 3
**Recommendation:** 6
**Confidence:** 3

**Main Review:**

Strengths:
- This paper utilizes several technologies to accelerate the branch-and-bound procedure on finding adversarial examples for deep neural networks.
- Compared to the baseline attack method (MIP attack), the proposed BaB attack achieves higher attack success rates within much less time.

Weaknesses:
- One drawback of the BaB attack is speed. Compared to the gradient-based attacks, which takes only a few seconds to generate adversarial example, the time cost of the BaB attack is not negligible.
- The BaB attack consists of several parts, including bean search guided by neural network verifiers, diving in branch and bound, and large neighborhood search. However, no ablation experiments were presented to verify the effect of each part.
- In the design of the BaB attack algorithm, many are heuristic and lack theoretical analysis. For example, the BaB attack utilizes lower bounds given by the bound propagation method $\alpha, \beta$-CROWN to guide beam search.
- The written is unclear and difficult to understand.

**Summary Of The Paper:**

This paper proposes a branch-and-bound attack (BaB-Attack) to solve hard instances efficiently, where none of the existing adversarial attacks can succeed. Specifically, the BaB-Attack utilizes the bound propagation-based neural network verifiers on GPUs to rapidly evaluate a large number of searching regions, builds an adversarial candidates pool to guide the search, and refines candidates adversarially
examples using a bottom-up large neighborhood search. Experimental results show that BaB-Attack outperforms the existing attacks in both attack success rates and efficiency.

**Summary Of The Review:**

Overall, I think this paper is marginally below the acceptance threshold.

---

> ### Author Response · Authors · 2021-11-22
> **Thank you for your great suggestions and we have added ablation experiments and greatly improved writing**
>
> Thank you for your constructive comments! We’ve greatly improved our paper by **adding new experiments on 4 more models and comparing to 5 additional attack baselines**, as well as  conducting **ablation studies** to support our algorithm design.
>
> We answer your questions in detail below:
>
> ### Ablation experiments, and results on new models and attack baselines
>
> We conduct detailed ablation experiments in Table 3. On two models, we disable part of our attack and investigate its performance in 5 different settings. Some conclusions we found are:
>
> - When the top-down beam search guided by the neural network verifier (α,β-CROWN) is disabled and random subdomains are selected instead, the attack performance significantly drops. So it is **important to use a neural network verifier to guide the search**.
>
> - When each part of our attack (diving, bottom-up search) is disabled, we find less adversarial examples. The optimal performance is obtained when all parts are enabled, especially on the harder CNN-A-Adv model.
>
> Additionally, we added **four new models** (CIFAR CNN-A-Adv, CIFAR CNN-A-Mix, CIFAR Marabou and MNIST Conv-Small) in Table 1 for comparison, and **5 strong adversarial attacks** (FAB attack, Square attack, MIFGSM attack, DIFGSM attack, Distributional attack) in Table 2. **All new results firmly support the strengths of our attack**: we can find many adversarial examples when none of these attacks succeed, and we are also much faster than the generic MIP solver based attack.
>
> ### BaB attack speed
>
> We do agree our attack is relatively slow because we conduct a systematic search on activation space. It should be used as a last resort after existing fast attack algorithms searching directly in input space (e.g., PGD) fail. In Table 2, we show that we can attack more images and discover adversarial examples where all existing adversarial attacks fail. Also, compared to some other strong attacks such as FAB and Square attacks, our time is similar to theirs (**see new results in Table 2**). Additionally, we are also significantly faster (>10x on most models) than the MIP based attack which is also an exhaustive attack on activation space like ours (Table 1).
>
> ### On theoretical analysis and the use of heuristics:
>
> Our attack algorithm is theoretically based on the mixed integer optimization problem in Eq. (2). This formulation allows us to enumerate every linear piece of a ReLU network and search for adversarial examples systematically. This is the main reason why we can find more adversarial examples than common attacks directly searching in the input space. We have added discussions on this in Intro and Background.
>
> Solving the attack formulation in Eq. (2) efficiently indeed needs a lot of heuristics (hardly with theoretical guarantees), just like solving integer programming (IP) or other NP-complete problems. A commercial solver such as Gurobi also has a lot of heuristics built-in. On these problems, a valuable algorithm should use heuristics to perform well on common problem instances, and it is hard to prove it is always effective since the target problem is NP-Complete (there will always be bad problem instances that take exponential time). In fact, much important progress of MIP solving is due to heuristics, and any efficient and practically useful heuristic is important (Achterberg and Wunderling, 2013).
>
> The importance of utilizing lower bounds obtained by an NN verifier such as α,β-CROWN is supported by our ablation study - without these lower bounds, the search is almost blind and random top-down search can only find very few adversarial examples.
>
> Finally, our paper is the **first work to demonstrate effective heuristics that are specialized for the adversarial attack problem and outperforms generic MIP solver** (MIP attack) by over an order of magnitude on many models (Table 1), so we do believe this is an important and novel contribution.
>
> ### Writing:
>
> We have significantly improved writing in our revision. Particularly, we have revised the motivations in Introduction to clearly state the drawbacks of existing attacks and the benefits of the systematic search of adversarial examples in activation space, and why directly using a generic MIP solver is ineffective. We also introduced better notations and formalism in the Methods section. We hope the reviewer can check out our paper again.
>
> Thank you again for all your comments. We really appreciate your feedback and hope to hear from you. **We will be very grateful if you can re-evaluate our paper based on the new ablation study and new revision of the paper**, and please kindly let us know if anything is still unclear to you.
>
>
> References:
>
> Achterberg, Tobias, and Roland Wunderling. "Mixed integer programming: Analyzing 12 years of progress." Facets of combinatorial optimization. Springer, Berlin, Heidelberg, 2013.

---

> ### Author Response · Authors · 2021-11-29
> **We hope the reviewer can check out our response and new results**
>
> Thank you again for your valuable comments. Since the discussion period is closing soon, we hope the reviewer can check out our [detailed response](https://openreview.net/forum?id=oxwsctgY5da&noteId=UqowN5vlVV0) and [the revised paper](https://openreview.net/pdf?id=oxwsctgY5da).
>
> In short, we have added a **detailed ablation study** in Table 3. Our ablation study shows that using lower bounds given by the bound propagation method α,β-CROWN is very important to guide the beam search. Additionally, we also included **extensive new experiments**: 4 more models in **Table 1**, and 5 more attack baselines in **Table 2**, and more plots in **Figure 6**. All new results strongly support the strength of our algorithm. We also revised part of the paper, especially the Introduction to make our **paper easier to understand and better motivated**.
>
> Regarding theoretical results, it can be hard to prove that beam search guided by α,β-CROWN is always helpful because it is an NP-complete problem and the worst case is always exponential. However, our heuristics are **well-motivated and consistently perform well** on many models. Like many adversarial attack papers, we believe good motivations for heuristics and strong empirical results are most important, and far-fetched or non-essential theoretical results can be distracting and misleading and are not helpful here.
>
> We hope the reviewer can reevaluate our work based on our new numerical results, our [detailed response](https://openreview.net/forum?id=oxwsctgY5da&noteId=UqowN5vlVV0) and the [new revision of the paper](https://openreview.net/pdf?id=oxwsctgY5da). Please kindly let us know if any of your questions have not been addressed, and thanks again for your insightful review.

---

> ### Comment · Reviewer_LiPk · 2021-11-29
> **Thanks for the detail responses**
>
> The authors addressed most of my concerns, thus I increased my score.

---

### Official Review · Reviewer_wXK6 · 2021-11-02

**Correctness:** 3
**Technical Novelty And Significance:** 3
**Empirical Novelty And Significance:** 3
**Recommendation:** 8
**Confidence:** 3

**Main Review:**

Strengths:
- The paper has good novelty, combining techniques from several other papers into an interesting new technique
- The proposed method is shown to provide a significant speed improvement over a vanilla MIP approach on the given tasks

Weaknesses:
- The experiments are only run on the adversarial examples which could not be solved by any other method, and only compared to the vanilla MIP approach. I would like to see some numbers comparing the proposed approach to some different methods for time and efficacy (the proposed method is slow so I would not expect to see this tried on a large dataset).


**Summary Of The Paper:**

This paper introduces an efficient way to apply a branch-and-bound procedure to solve the mixed integer programming formulation of the problem of creating adversarial examples. This method of adversarial example creation is particularly designed to work in difficult cases where other methods fail.

**Summary Of The Review:**

Overall the paper is good, demonstrating a novel method to quickly produce adversarial attacks in hard cases. It could be improved with the inclusion of more extensive experiments.

---

> ### Author Response · Authors · 2021-11-22
> **Thank you for the encouraging comments, and we have added extensive experiments as you suggested**
>
> We thank the reviewer for the encouraging comments and suggesting a comparison to other adversarial attacks. We have provided **extensive experiments in our revision**:
>
> 1. We have conducted a comprehensive experiment on **5 more strong adversarial attacks** on **9 MNIST and CIFAR-10 models** in Table 2.
>
> In Table 2, we conduct attacks on the **hard instances** listed in Table 1, which cannot be attacked by multi-targeted PGD attack, Output Diversified Sampling PGD attack (both are stronger than vanilla PGD attack) as well as AutoAttack. We report both the number of successful attacks and average time. The hard instances are indeed very hard, and many attacks in Table 2 fail to find any adversarial examples. Some attacks like Square and FAB attacks can find some adversarial examples on this hard instances but are **much fewer than ours** and they are also **slow**.
>
> 2. We also add more extensive evaluation:
>
> In Table 1, we added **3 additional CIFAR-10 models** (CNN-A-Mix and CNN-A-Adv from Dathathri et al., 2020; Marabou from (Bak et al., 2021)) and **one additional MNIST model** (ConvSmall from Singh et al., 2018a). Our method performs well on all the newly added models: we are over an order of magnitude faster than MIP attack, and also produce more adversarial examples.
>
> In Figure 6, we include **3 new plots** of attack margins, for the newly added CIFAR CNN-A-Mix, CIFAR Marabou, and MNIST ConvSmall models. On all three models, we achieve smaller margins between the ground-truth class and class under attack (other baselines are under the y=x lines), indicating our attack is stronger.
>
> 3. We include an **ablation study** to see the importance of each part of the algorithm in Table 3. We found that the beam search guided by a neural network verifier is the most important contributor, while other tactics (diving, bottom-up search) are also helpful.
>
> We hope all the new empirical results are helpful for addressing your concerns, and we will really appreciate it if you can reevaluate our paper based on these new results, and let us know if you have further questions. Thank you!

---

> ### Author Response · Authors · 2021-11-29
> **We hope the reviewer can check out our extensive new experiments and the revised paper**
>
> Thank you again for your encouraging and constructive review! Since the discussion period is closing soon, we hope the reviewer can check out the **extensive experiments** we added [in the revision](https://openreview.net/pdf?id=oxwsctgY5da) and [our detailed response](https://openreview.net/forum?id=oxwsctgY5da&noteId=gTDzePzX8qo). Particularly, we **added 5 more strong adversarial attack baselines** as you suggested in Table 2, and also added **4 additional models** in Table 1, **new plots** in Figure 6, and an **ablation study** in Table 3. All new results strongly support the strengths of our algorithm.
>
> We sincerely hope the reviewer can check out the new results in [our revision](https://openreview.net/pdf?id=oxwsctgY5da) and reevaluate our paper, and please kindly let us know if you have any further questions. Thank you!

---

> ### Comment · Reviewer_wXK6 · 2021-11-29
> **Increasing my score due to paper improvements**
>
> The authors have done a good job of addressing my concerns with the paper therefore I have increased my score.

---

### Official Review · Reviewer_G18n · 2021-11-03

**Correctness:** 3
**Technical Novelty And Significance:** 3
**Empirical Novelty And Significance:** 2
**Recommendation:** 8
**Confidence:** 2

**Main Review:**

This paper is neatly written, the intuition and heuristic method is easy to follow, and experiments are convincing.
However, this paper is purely heuristic-based where some theoretical results would be much appreciated, and experiments are not comprehensive, I feel like by keep working towards either one of these two aspects will highly improve the quality of this paper.
Also, while the adversarial attack is a very important problem, I think the basic idea presented in this paper is not fundamentally novel and in that regard, the contribution of this paper seems a bit weak.
In the following I list some of the detailed comments.

1. Throughout, "branch-and-bound" and "branch and bound" are both used. You should be consistent, and the first one is more commonly used. Also, its abbreviation is usually denoted as B&B.
2. Line after equation (1): "ball around input x", here $x$ should be $x_0$, denoting the benign input point.
3. When you mention "projected gradient descent" you should mention (PGD) as its abbreviation, since you used PGD heavily in the later context.
4. You should specify that N is the number ReLU neurons here.
5. Second paragraph, page 2: You should specify that $N$ is the number ReLU neurons here.
6. Second line on page 3: "adversarial adversarial attack" is wrong.
7. All Eq. \ref{...} should be Eq. \eqref{...}
8. For Eq. (2), you should mention for i=1, ..., L. Also, I think you should also impose $\hat{z}^{(i)}(x) \geq 0$.
9. Page 3: "can be provable obtained" should be "... provably ..."
10. The end of page 3: "... is well explored recently ..." here a reference would be appreciated.
11. "Adversarial common pattern" on page 6 should be highlighted in italic.
12. Second paragraph of page 7: "T percent of" doesn't make sense, it should be "100*T percent".
13. In Fig. 6, for the experiments in (c), I cannot see the obvious improvement for attack margin.
14. Detailed information of those data-sets should be mentioned.
15. In Experiment section, it's better to mention Fig. 6 somewhere in this section, while giving a more detailed explanation.
16. Experiments can be carried out and presented in a more detailed fashion.

**Summary Of The Paper:**

In this paper, the authors propose to solve the adversarial attack problem of ReLU neural network by using a branch-and-bound procedure, and search adversarial examples within the activation space corresponding to binary variables of a mixed-integer programming formulation. The experiments showcase that such adversarial attack framework enables them to find adversarial examples for hard instances where existing adversarial attacks fail.

**Summary Of The Review:**

This paper can be much improved by conducting a more extensive numerical experiments, basic idea is obvious and not fundamentally novel.

---

> ### Author Response · Authors · 2021-11-22
> **Thank you for the very detailed feedback and we have added extensive numerical experiments (1/2)**
>
> We greatly appreciate the very detailed comments by the reviewer! It’s very helpful for us, thank you! We have fixed all the typos and minor issues you mentioned, and also significantly improved the writing of our paper. Particularly, we rewrote the introduction to make the contribution more clear, and also added **extensive experiments on 4 new models and comparison to 5 new attack baselines**, as well as an **ablation study experiment**.
>
> ### 1. We have added extensive numerical experiments, detailed below:
>
> 1. We added **three additional CIFAR-10 models** (CNN-A-Mix and CNN-A-Adv from Dathathri et al., 2020; Marabou from (Bak et al., 2021)) and **one additional MNIST model** (ConvSmall from Singh et al., 2018a). Our method performs well on all the newly added models: we are over an order of magnitude faster than generic MIP solver based attack (MIP attack), and also produce more adversarial examples.
>
> 2. We added **5 additional strong attacks** into comparison in Table 2. We run all 5 attacks (FAB attack, Square attack, DIFGSM, MIFGSM, and Distributional attack) on the hard instances (listed in Table 1) on a total of **9 models**. These are indeed very hard instances and many attacks fail to find any adversarial examples. Some existing attacks like Square and FAB attack can find some adversarial examples, but are much fewer than ours and they are also slow.
>
> 3. In Figure 6, we include **3 new plots** of attack margins, for the newly added CIFAR CNN-A-Mix, CIFAR Marabou, and MNIST Conv-Small models. On all three models and all tested images, we achieve the smallest margins between ground-truth class and attack target class (other baselines are under the y=x lines).
>
> 4. We include an **ablation study** to see the importance of each part of the algorithm in Table 3. We found that the beam search guided by a neural network verifier is the most important contributor, while other tactics (diving, bottom-up search) are also helpful.
>
> We hope the extensive experiments on new models and new attacks can address your concern on no comprehensive experiments.
>
> ### 2. On the theoretical side, and the use of heuristics
>
> Our attack algorithm is theoretically based on the mixed integer optimization problem in Eq. (2). A theoretical advantage of this formulation is that it allows us to enumerate every linear piece of a ReLU network and search for adversarial examples systematically. This is the main reason why we can find more adversarial examples than common attacks directly searching in the input space. We have added discussions on this in Intro and Background.
>
> Solving the attack formulation in Eq. (2) efficiently indeed needs a lot of heuristics (hardly with theoretical guarantees), just like solving integer programming (IP) or other NP-complete problems. A commercial solver such as Gurobi also has a lot of heuristics built-in. On these problems, a valuable algorithm should use heuristics to perform well on common problem instances, and it is hard to prove it is always effective since the target problem is NP-Complete (there will always be bad problem instances that take exponential time). In fact, much important progress of MIP solving is due to heuristics, and any efficient and practically useful heuristic is important (Achterberg and Wunderling, 2013).
>
> Our paper is **the first work to demonstrate effective MIP heuristics that are specialized for the adversarial attack problem** and outperforms a generic MIP solver by an order of magnitude (Table 1) on many models, so we do believe this is an important and novel contribution.

---

> ### Author Response · Authors · 2021-11-22
> **Thank you for the very detailed feedback and we have added extensive numerical experiments (2/2)**
>
> ### 3. On contribution and novelty:
>
> We apologize for not stating our novelty clearly in our initial submission. To make our contributions more clear, we rewrote the introduction section. Let’s summarize it here:
>
> Our work aims to find adversarial examples on activation space with 0-1 variables, which allows a **systematic search of adversarial examples using branch and bound**. Existing attacks searching in the input space (e.g., via gradient descent) cannot exhaustively search for adversarial examples in the continuous and high dimensional input space.
>
> Although using a MIP solver can also search in activation space, **a generic MIP solver can be very inefficient** because it cannot exploit the structure of the network to speed up solving, cannot utilize information from cheap adversarial attacks, and also can hardly be accelerated on GPUs. So although attacks on activation space can be powerful, they are extremely slow using a generic MIP solver.
>
> Our work developed **a specialized search algorithm (Section 3) to overcome these weaknesses**. We use bound propagation methods to exploit the structure of the networks to speed up solving; we maintain a pool of existing adversarial examples to guide the search on activation space; our search is also accelerated on GPUs. Overall, we achieved an order of magnitude speedup on most models compared to a generic MIP solver for attacks, and can discover adversarial examples where none of existing attacks can succeed.
>
> We’ve rewritten the Introduction to make our contribution and novelty more clear, and we hope the reviewer can check out our revision.
>
> ### 4. Other minor questions:
>
> 1. "branch-and-bound" and "branch and bound" are both used
>
> We’ve unified the notations to "branch and bound" and "BaB". We use "BaB" because several early papers on neural networks verification (such as Bunel et al., NeurIPS 2018 and many follow-up works) used this abbreviation.
>
> 2. Cannot see improvements in Fig. 6(c)
>
> We were not able to finish running all examples in Fig. 6(c) before submission. Now we have updated Fig. 6(c) with more examples, and the results should be more consistent. Additionally, we’ve added Fig. 6(d)(e)(f) to show the effectiveness of our methods on three new models.
>
> **Thank you again for the very constructive and helpful comments, and we hope the reviewer can re-evaluate our paper based on the extensive experiments added in the revision**, and we hope we have addressed your concerns. Please kindly let us know if you have any further questions for us.
>
>
> References:
>
> Achterberg, Tobias, and Roland Wunderling. "Mixed integer programming: Analyzing 12 years of progress." Facets of combinatorial optimization. Springer, Berlin, Heidelberg, 2013.

---

> ### Comment · Reviewer_G18n · 2021-11-27
> **Keep my original score**
>
> I think the authors have carefully addressed most of my concerns, and I do appreciate the efforts been put into those new experiments. At the same time I am not fully convinced by the authors' justification about the loss of theoretical analysis. Overall, I decide to keep my original score unchanged.

---

> > ### Author Response · Authors · 2021-11-28
> > **Thank you for your feedback and follow-up discussions regarding theoretical analysis**
> >
> > Thank you very much for the follow-up and we are glad to hear most of your concerns are addressed. Regarding the theoretical analysis, we truly believe it is not very useful and can be distracting due to the nature of this paper; we believe far-fetched or non-essential theoretical results are not really helpful in a good paper.
> >
> > **Most existing attack papers including widely accepted ones like CW attack (IEEE S&P 2017) [Ref A], Multi-Targeted PGD (arXiv 2019) [Ref B], ODS-PGD (NeurIPS 2020) [Ref C], and AutoAttack (ICML 2020) [Ref D] mainly focus on insights of their designs and empirical results without providing many theoretical results**. This is partly because the attack problem itself is an NP-hard problem at worst case (solving the exact solution in Eq. (1)) [Ref E]; In practice, only approximations can be provided, and good heuristics must be developed to conduct attacks efficiently. It can be hard to develop useful and meaningful theoretical results for these heuristics, so most papers focus on demonstrating comprehensive empirical results instead.
> >
> > On the other hand, our paper can also be seen as specialized heuristics for solving the MIP problem of adversarial attacks. Theoretical analysis is usually not provided (and is often not practical nor very useful) for widely known heuristics in generic MIP problems, such as diving and local search heuristics [Ref F-L]. Useful heuristic insights along with strong empirical results should be sufficient to demonstrate their strengths and contributions, and these heuristics are very important in mature MIP solvers for solving real-world problems.
> >
> > In our paper, we provided detailed insights on why each proposed heuristics can help in Section 3 and also extensive empirical results in Section 4 to demonstrate the efficacy of our proposed heuristics. Especially in the ablation study (Table 1), we have shown the success and importance of each proposed method, supporting the insights we provided. As an empirical attack paper, we think the empirical results are strong enough to demonstrate the contributions of our proposed methods.
> >
> > One theoretical result we could have added is to show the completeness of our search procedure, based on the discussion on “Completeness” in the middle of page 5. However, as we discussed in this paragraph, achieving completeness is hardly possible in practice, and we feel it can be misleading and distracting for readers. If the reviewer feels this result is useful, we can certainly rewrite part of this paragraph as a theorem (its proof is immediate based on Zhou & Hanson 2005).
> >
> > Thank you again for your constructive review and helpful discussions. Please kindly let us know if you have further concerns regarding the necessity of theoretical results. We sincerely hope our explanation could clarify your concerns, and we are very grateful for the discussions.
> >
> > References:
> >
> > [Ref A] Nicholas  Carlini  and  David  Wagner, Towards  evaluating  the  robustness  of  neural  networks, IEEE Security and Privacy (SP) 2017.
> >
> > [Ref B] Sven Gowal, Jonathan Uesato, Chongli Qin, Po-Sen Huang, Timothy Mann, and Pushmeet Kohli, An alternative surrogate loss for pgd-based adversarial testing, arXiv preprint arXiv:1910.09338, 2019.
> >
> > [Ref C] Yusuke Tashiro, Yang Song, and Stefano Ermon. Diversity can be transferred: Output diversification for white-and black-box attacks, NeurIPS 2020.
> >
> > [Ref D] Francesco Croce and Matthias Hein, Reliable evaluation of adversarial robustness with an ensemble of diverse parameter-free attacks, ICML 2020.
> >
> > [Ref E] Guy  Katz,  Clark  Barrett,  David  L  Dill,  Kyle  Julian,  and  Mykel  J  Kochenderfer, Reluplex:  An efficient smt solver for verifying deep neural networks, CAV 2017.
> >
> > [Ref F] Timo Berthold, Primal heuristics for mixed integer programs, 2006.
> >
> > [Ref G] Egon Balas, Clarence Martin, Pivot-and-complement: a Heuristic for 0-1 programming, Manage Sci 1980.
> >
> > [Ref H] Matteo Fischetti, Fred Glover, Andrea Lodi, The feasibility pump, Math Program 2005.
> >
> > [Ref I] Pierre Hansen, Nenad Mladenovic, Dragan Urosevi, Variable neighborhood search and local branching, Comput Oper Res 2006.
> >
> > [Ref J] Edward Rothberg, An evolutionary algorithm for polishing mixed integer programming solutions, Informs J Comput 2007.
> >
> > [Ref K] Timo Berthold, RENS - relaxation enforced neighborhood search, Technical Report 2007.
> >
> > [Ref L] Vinod Nair, et al., Solving mixed integer programs using neural networks, arXiv preprint arXiv:2012.13349, 2020.

---

> > > ### Comment · Reviewer_G18n · 2021-12-01
> > > **Increase my score**
> > >
> > > Thank you for your detailed explanation. Since now I am more convinced by your arguments, I decide to raise my score.

---

### Author Response · Authors · 2021-11-24
**General Response: Extensive New Experiments and Revisions in Our Paper**

Dear Reviewers,

Thank you again for providing insightful reviews for our paper. As suggested by the reviewers, we have added significantly more experiments and also rewrote part of the paper for presenting our motivations and algorithm better. Our paper has become much stronger now.

Reviewer G18n, wXK6, and LiPk mentioned our contribution can be enhanced by adding more experiments, and in our revision, we have added **comprehensive new experiments**:

1. **Four more new models**: In **Table 1**, we added three additional CIFAR-10 models (CNN-A-Mix and CNN-A-Adv from Dathathri et al., 2020; Marabou from (Bak et al., 2021)) and one additional MNIST model (ConvSmall from Singh et al., 2018a). Our method performs well on all the newly added models: we are over an order of magnitude faster than generic MIP solver based attack (MIP attack) on many models, and also produce more adversarial examples.

2. **Five additional strong attack baselines**:  In **Table 2**, we run 5 additional strong adversarial attacks (FAB attack, Square attack, DIFGSM, MIFGSM, and Distributional attack) on the hard instances (listed in Table 1) on a total of 9 models. The hard instances were filtered using PGD and AutoAttack, and are indeed very hard: many attacks fail to find any adversarial examples. Some attacks like Square and FAB attack can find some adversarial examples but are much fewer than ours and they are also slow.

3. **Ablation study**: In **Table 3**, we disable parts of our algorithm to see the importance of each part. We found that the beam search guided by a neural network verifier is the most important contributor, while other tactics (e.g., diving, bottom-up search) are also helpful.

4. **New figures**: In **Figure 6**, we include plots of attack margins, for  3 more models: CIFAR CNN-A-Mix, CIFAR Marabou, and MNIST Conv-Small. On all 3 models and all tested images, we achieve the smallest margins between the ground-truth class and attack target class (other baselines are under the y=x lines), indicating our attack is consistently stronger. In **Figure 8** (in appendix), we include the number of attacked images vs. time comparison on 5 more models, and we significantly outperform the existing MIP solver (Gurobi).

Additionally, as suggested by Reviewer aJSd and LiPk, we have also greatly improved our writing:

1. **We rewrote the Intro for better motivation**:  In Intro, we now clearly state the drawbacks of existing attacks, the benefits of the MIP formulation, why directly using a generic MIP solver is ineffective, and how we tackle the weaknesses in a MIP solver. In short, attacks based on searching the input space (e.g. PGD) cannot search systematically and may miss adversarial examples due to non-convexity. The MIP formulation allows us to systematically search the activation space and find more adversarial examples on hard instances. However, using a generic MIP solver can be slow and ineffective. We propose specialized top-down beam search and bottom-up search procedures, which utilize the structure of the network via bound propagation to solve the problem more efficiently, use cheaply generated adversarial examples to guide the search, as well as enable GPU acceleration (none of these are possible in existing MIP solvers). Overall we achieve >10x speedup on many models and also find more adversarial examples on hard instances.

2. **We also introduced better notations and formalism** in the Methods section. Especially, we now use formal math equations to accurately describe some crucial parts of our procedure.

Finally, we have addressed the questions of each reviewer in individual replies. We hope reviewers can check out the new version of our paper and our response, and kindly reevaluate our paper. If there is still something unclear, please kindly let us know. Thank you.

Sincerely,
Paper 2303 authors

---

### Decision · Program_Chairs · 2022-01-20

**Decision:**

Reject

**Comment:**

This work proposes a branch and bound framework for adversarial attacks, based on the MIP formulation of attacking against ReLU networks. It adopts several heuristic tricks to accelerate the attack efficiency, and shows better attack performance on hard examples, compared to off-the-shelf MIP solvers.

The main concerns in the first-round reviews include:
1. The limited novelty, since the problem and formulation is not novel, and the proposed method is the combination of several heuristic tricks, without theoretical analysis.
2. The experiments are insufficient, and only MIP solvers are compared, while other attack methods and the ablation study of different tricks are not presented.
3. The efficiency is higher than off-the-shelf MIP solvers, but significantly lower than other adversarial attack methods.

The authors made great efforts to respond these concerns, such as adding some baseline attack methods and ablation studies. Most reviewers appreciated the authors' efforts and raised their initial score. However, after reading the revised manuscript, reviews and responses, I think many serious issues still exist.
1. Since the proposed method is only applicable to the MIP formulation of attacking ReLU networks, it could not become a new baseline to attack mainstream deep neural networks, which significantly restrict its practical contribution to the community. And, it didn't provide novel theoretical tools or better theoretical results of analyzing the robustness of ReLU networks. It just reduce the gap to the verified robustness of existing verification methods. Thus, I cannot find significant contributions to the community of adversarial examples, from both practical and theoretical perspectives.
2. The experiments are still insufficient. There have been massive advanced white-box and black-box attack methods. The added FAB, square (black-box), DIFGSM, MIFGSM (white-box), are popular methods, but not SOTA methods. Besides, only small images of MNIST and CIFAR-10 are tested, while large-scale images like ImageNet are not tested. It has been observed in many works, and according to my own experience, the attack performance on between small images (MNIST and CIFAR-10) and large images (ImageNet) is not always consistent. And, the low efficiency weakness (compared to other attack methods) of the proposed method may be further highlighted.

It is not easy to decide to reject a submission with average score 7. The authors's efforts during their submission and the responses are greatly appreciated. However, I would like to accept works that can bring in real contributions to the research community. Hope the reviews and meta review could be helpful to further improve this work.